# Assessment of Some Clay-Based Products Available on Market and Designed for Topical Use

Carla Marina Bastos [1,2,*] and Fernando Rocha [1]

1    GeoBioTec, Department of Geosciences, University of Aveiro, 3810-193 Aveiro, Portugal
2    Exatronic, Lda. 3800-373 Aveiro, Portugal
*    Correspondence: mbastos@exatronic.pt; Tel.: +351-234-315-500

**Abstract:** The establishment of quality requirements of clay-based products, for medicinal, wellness, and aesthetic purposes, is mainly sustained by the good interactions between the clay-based formulation and the skin. The release of ionizable elements and their availability to percutaneous absorption should be, ideally, physiologically effective during passive percutaneous absorption. Clay-based products are promoted in the European market as therapeutic clays or aesthetics, which is labeling that combines characteristics of medicinal products along with cosmetics. Different countries regulate these products under different legal frameworks. This study focuses on the mineralogical, chemical, and technological characterization of some clay-based products available on the market, designed for topical use, framed in the peloids concept, and claimed as natural products. The main goals are to contribute to the establishment of clay-based products quality criteria as reliable scientific information, aiming for the compliance of intended use, the information for the potential health hazards and toxicological effects of clay-based products, and the distinction in what concerns therapeutic compliance and aesthetic or wellbeing product certification. There were 13 clayed products for cosmetic purposes, available online and in commercial stores, together with three thermal peloids, that were studied. Mineralogical composition of the 16 studied samples reveals a polymineralic association with the presence of variable quantities of quartz, calcite, and feldspars, whereas clay minerals are not predominant and characterized by the presence of clay-based fraction content, composed mainly by illite, smectite, and kaolinite in variable amounts and with several mineral associations. The clay-based products contain median values of 17 ppm As, 315 ppm Ba, 79 ppm Cr, 11 ppm Co, 29 ppm Pb, 26 ppm Ni, and 62 ppm Zn. One sample presented 4.1 ppm of Cd. The studied samples have safety concerns about specific limits of As, Ba, Cd, Cr, Co, Pb, Ni, and Zn which are above the regulated avoidable limits. Samples' pH is out of range of skin's natural pH as well.

**Keywords:** pelotherapy; peloids; medical geology; cosmetics; transdermal

## 1. Introduction

Clays for medicinal, wellness, and aesthetic purposes were ancestrally used by mankind topically (peloids or muds) or by ingestion [1].

Peloids, resulting from the mixing of clay and mineral–medicinal water, have been used in many thermal centers, more recently in designated spas, as a distinguished therapeutic modality, pelotherapy. All over the world, we can find different peloids that can be highlighted by their peculiar chemical and mineralogical composition, physical proprieties, and also by the biological–metabolic activity of micro-organisms, e.g., Poça da Dona Beija (Portugal—S. Miguel Island, Azores), Porto Santo (Portugal—Madeira Island), Dax peloid (France), Copahue spa (Argentina), Arnedillo, Caldas de Boí, and El Raposo spas (Spain), Abano–Montegrotto spas (Padua, Italy), among others.

The majority of these peloids are naturally maturated with the autochthonous water and clays on the outskirts of thermal centers, or they are prepared artificially (para-mud), sometimes enhanced with cosmetic ingredients performers—botanicals, algae, or

diatoms—as well as colored additives or flavors. Some of these peloids are sold for cosmetic purposes and personal healthcare usage, a parallel business to the gist of pelotherapy, and are now gaining some relevance in the wellness field and in Health Tourism.

The heated muds, when in contact with the skin, promote several reactions responsible for the therapeutic effect of the clay essential elements [2]. Several clay-based products (peloids) have a traditional therapeutic history, being easily available in nature for people use [3–6] or are available on the market as 100% natural products [7]. Many of these products have no pharmaceutical or cosmetic control, and they are not free of possible side effects on human health [8,9].

The pharmaceutical and cosmetic industry uses clays in their formulations that are subject to prior control before being used [10,11]. Clays are used in pharmaceuticals, as excipients or as active substances, due to their high retention capacity, colloidal and expansive properties, usefulness for the modulation of drug release in the organism [12], chemical inertia, and low or non-toxicity to the patient [13].

Pelotherapy's studies emphasized the suitable properties of peloids when applied to the skin, such as water retention capacity, consistency, adhesiveness, heat capacity, cooling rate, cation exchange capacity, handling, and pleasant sensation [14–16]. The focus on the thermotherapeutic effect of the peloid as having an important healing action, by the heat released when in contact with the skin, is still a transversal property for the characterization of peloids for therapeutic purposes [17]. The importance of not neglecting the mobility of some chemical elements with toxicity potentiality after their absorption by the skin [9,13,18] is now considered with strong scientific relevance, which is framed by the legal conformity associated with consumer safety and health users. Contemporaneous studies found evidence of the need for quality criteria establishment and certification of clay-based products intended to be used topically, namely thermal muds, which have therapeutic action and are applied directly onto the skin on thermal centers [15,16,19], as well as the proposal of some methodologies for clays' decontamination before their incorporation into cosmetics, to achieve the limits required for cosmetic safety [20].

The requirements' definitions for good interaction between the clay-based formulation and the skin are sustained on the increased release of ionizable elements and their disposal to percutaneous absorption. The results should, ideally, be physiologically effective during passive percutaneous absorption [18]. The solubility, molecular mass, depth of penetration, and toxicology of the clay components while a topic vehicle need to be considered in the percutaneous absorption. The cation exchange capacity (CEC) and the other formulation characteristics may define the percutaneous depth efficacy that ions may reach, as well as the desirable absorption by the skin. The success of a complete percutaneous action can be observed by the pain relief, the anti-inflammatory action, the range improvement of the upper limb movements, the antibacterial action, the healing action [21], and others therein. These clay-based products are mainly used in rehabilitation programs at thermal centers and spas, being associated with musculoskeletal and tendon injuries, rheumatic pathologies, dermatological infirmities, or for aesthetic purposes and skincare. The biological and physiological mechanisms of how mud applications alleviate symptoms of several pathologies, in the dermatological and rheumatological field, are still not completely understood [22–27].

Considering the current European regulatory and legal framework on cosmetic products, namely the EC No. 1223/2009, it is observed that it is the nearest compliance guideline for the quality criteria establishment of peloids or clay-based products. The Directive 76/768/EC, which was adopted in 1976, was replaced by Regulation (EC) No 1223/2009 from 2013 onwards due to the many amendments made to it and the new amendments that were required. According to the actual European Regulation on cosmetic products, a cosmetic product is defined as "any substance or mixture intended to be placed in contact with the external parts of the human body with the main, or exclusive aim, of cleansing or perfuming it, changing their appearance or smell, as well as protecting and maintaining it". We can find a similar definition by the Federal Food, Drug, and Cosmetic Act, 2021 [28];

however, the labeling requirements of the same product, regulated as cosmetic in Europe, could be regulated as a drug in the United States. The term sunscreen or sun protection meets the definition of a cosmetic product by the European regulation, whereas the FDA considers it as a drug because it is expected to be a product for skin protection from the harmful effects of the sun. In Europe, the legislation on medicinal products is not fully harmonized and may be classified under national regulations. The safety evaluation of cosmetics, in what concerns the chemical content, is regulated by the provisions of EC No. 1223/2009, the EU REACH Regulation (EC No. 1907/2006), Commission Regulation on claims in cosmetics products (EU No. 655/2013), and with national laws where cosmetics products will be on market. However, some products are defined as borderline products when it is unclear whether a product is a cosmetic within the definition in the cosmetic regulation or whether it falls under other sectorial legislation [29]. Nevertheless, from a technological standpoint, there is no regulation or criteria established either for clay-based cosmetics or for peloid formulations.

Viseras et al. (2022) debated the European cosmetic regulation, supported by the rules of the intended use and by the different categories and typologies, according to the European cosmetic directive 76/768/ECC list in annex I and the European association of cosmetics fabricants. The use of clays and derivates as ingredients were used in numerous formulations for commercial cosmetics products with both technological and cosmetological functions [28].

Some clay-based products (zeolites and bentonite-montmorillonite) are presented in the market as natural medicines for ingestion, with a detoxification action and a protective effect on the mucous membrane of the gastrointestinal tract, and they are certified as medical devices.

The European legal framework governing medical devices, Regulation (EU) 2017/745, defines a medical device as "any instrument, apparatus, appliance, software, implant, reagent, material or other article intended by the manufacturer to be used, alone or in combination, for human beings for one or more specific medical purposes". This medical device regulation is much closer to that of the FDA prerequisites for the quality conformity assessment. European Regulation and FDA requires the determination of a product as a medical device to be settled on the intended use and indications for use. After that, it should be verified if it meets the medical device definition. The definition of a medical device by the FDA consists as "an instrument, apparatus, implement, machine, contrivance, implant, in vitro reagent, or other similar or related article, including a component part or accessory which is: (i) recognized in the official National Formulary, or the United States Pharmacopoeia, or any supplement to them; (ii) intended for use in the diagnosis of disease or other conditions, or (iii) in the cure, mitigation, treatment, or prevention of disease, in man or other animals, or (iv) intended to affect the structure or any function of the body of man or other animals, and which does not achieve its primary intended purposes through chemical action within or on the body of man or other animals and which is not dependent upon being metabolized for the achievement of any of its primary intended purposes."

This study aims at the mineralogical, chemical, and technological characterization of some clay-based products available on the market, designed for topical use, framed in peloids concept, and claimed as natural products. The main goals are to yield comparative data that can contribute to the establishment of specific quality parameters and clarify the effect of the differentiated usage of peloids.

## 2. Materials and Methods

### 2.1. Materials Description

There were 13 clayed products for cosmetic purposes, available online and in commercial stores, together with three thermal peloids (C5, C10, and C16), that were studied. There were eight samples obtained in a semi-solid formulation (paste), and the remaining samples were acquired in the form of powder. Despite sample C7, which is for ingestion, all samples are indicated for dermal application, generally for face masks, cataplasm, or

whole-body use (Tables 1 and 2). The sample C7 is sold as a medical device. The samples C1, C2, C3, C4, C6, and C8 were formulated with mineral thermal water.

　　The paste samples are specially designed as peloids with preparations 'ready-to-use'. The labeled indications/information commonly included healing, cleanse, detox, absorbent, refreshing, calming, decongesting, and energizing actions. However, we can find that there is no harmonized labeled information for the usage and dosage indication. The expected adverse events are centralized in skin reactions, and the safety alerts are based on eye and mouth avoidance.

**Table 1.** Label composition information of the studied samples.

| Commercial ID | Country | Composition |
|---|---|---|
| C1 [2] | Spain | AQUA (mineral medicinal water) bentonite, aloe barbadensis extract, menta piperita extract, arnica montana. |
| C2 [2] | Spain | AQUA (mineral medicinal water), bentonite, aloe barbadensis extract, menta piperita extract. |
| C3 [2] | Spain | AQUA (mineral medicinal water), bentonite, aloe barbadensis extract, calendula officinalis extract, menta piperita extract. |
| C4 [2] | Israel | Silt (Dead Sea mud), Aqua (Mineral Thermal water), Maris Slat (Dead Sea Salt), Phenoxyethanol, Caprylyl Glycol, Chlorphenesin. |
| C5 [2] | France | Controlled combination of muds from L'Adour river, mineral water and biological ingredients (Clostridium bifermentans and Cyanobacteria) |
| C6 [2] | Italy | Solum Fullonum (Fuller's earth) AQUA Termale di Abano, phenoxyethanol, sodium dehydroacetate, ethylhexyglycerin, citric acid |
| C7 [1] | Hungary | Zeomineral Products 100% natural mineral (zeolite)Internal use. |
| C8 [2] | Argentine | Bentonite, volcanic sediments, Kauline, Petrolatum Glycerin, Cetilic alcohol, isopropyl miristate, Triethanolamin, Etoxilaed lanolin, Polysorbate 20, Carbopol, Pentaglycan, Methylparaben, Propylparaben, Parfum, Deionized water. |
| C9 [1] | France | Aqua, Kauline, illite, montmorillonite, propanediol, glyceryl Undecylenate (natural origin ingredients) |
| C10 [2] | Spain | - |
| C11 [1] | Spain | Green Montmorillonite |
| C12 [1] | Spain | Yellow Montmorillonite |
| C13 [1] | Spain | Kaoline |
| C14 [1] | Spain | Red Illite |
| C15 [1] | Spain | Ghassoul (Morocco) |
| C16 [2] | Portugal | - |

[1] Powder. [2] Paste.

**Table 2.** Label indications for use of the studied samples.

| ID | Label Indications for Use | | | | | Therapeutic or Aesthetic Action |
|---|---|---|---|---|---|---|
| | Temperature | Application Dose | Action Time and Procedure | Periodic Application | Adverse Effects Caution | |
| C1 | 2 min/43 °C (Microwave) 15 min (double boiler method) | Not quantified ("enough quantity") | Leave 20 min covered with transparent film. Remove simultaneously with film, clean with water and dry. | 2–3 times a week. Recommended daily application | - | Antioxidant action on the skin and other body parts. It effectively neutralizes and combats the damage caused by the action of free radicals that cause aging, causing a barrier effect on them and consequently preventing and improving their symptoms. Muscle relaxant (relief from contractures). |
| C2 | 2 min/43 °C (Microwave) 15 min (double boiler method) | Not quantified ("enough quantity") | Leave 20 min covered with transparent film. Remove simultaneously with film, clean with water and dry. | 2–3 times a week. Recommended daily application | - | Antioxidant action on the skin and other body parts. It effectively neutralizes and combats the damage caused by the action of free radicals that cause aging, causing a barrier effect on them and consequently preventing and improving their symptoms. Muscle relaxant (relief from contractures). |
| C3 | Cold application | Not quantified ("enough quantity") | Leave 15 min covered with transparent film. Remove simultaneously with film, clean with water and dry. | 2–3 times a week. Recommended daily application | - | Effective and efficiency action in the immediate and prolonged relief of discomfort by feeling tired and heavy legs, preserving the biological venous structures. Delays the appearance of varicose veins. Antioxidant action on the skin and other body parts. It effectively neutralizes and fights the damage caused by the action of free radicals that cause aging, causing a barrier effect on them. Emollient, exfoliating, revitalizing, sebum regulator, keratolytic and keratoplastic effect. |
| C4 | Cold application and Microwave or double boiler method— 2 min/medium temperature | Not quantified ("spread generously on the skin") | Leave 5 to 10 min and rinse well with water. To relieve sore muscles and joints, heat the package | - | Avoid open wounds and irritated skin areas | The Black Mineral Peloid acts in depth, cleansing, purifying and restoring the skin's natural moisture balance for a smooth, radiant, revitalized look [30]. |

**Table 2.** *Cont.*

| ID | Label Indications for Use | | | | | Therapeutic or Aesthetic Action |
|---|---|---|---|---|---|---|
| | Temperature | Application Dose | Action Time and Procedure | Periodic Application | Adverse Effects Caution | |
| C5 | Hot application | - | - | - | - | Musculoskeletal disorders [1,31] |
| C6 | Cold application | Mix 1/3 of clay (500 g) and 1/3 of colored clay (200 g). | Body clay massage for 10 min. Leave 15 min for clay action and then clean and relax 10 min in a hot bath. At the end hydrate the body with thermal water and/or Abano Spa body cream with a massage. | - | - | Dermatological beneficiation. Antiage protection; antiage intensive; body tonic, body slim and microscrub effect. Musculoskeletal disorders [22,30,32,33] |
| C7 | - | - | - | - | - | Treatment of diarrhea, abdominal pain and heartburn. For the treatment and prevention and relief of food allergies, food intolerance, enteric infections and mild food poisoning. Medical supplement for additional treatment and reduction in symptoms of chronic digestive disorders, irritable bowel syndrome (IBS), ulcerative colitis and cholecystopathy (gallstone). |
| C8 | Cold application | - | Applied as a face mask for 10–15 min. Clean with water. | 2 times a week. Perfervidly by the morning and after bath. | Avoid eyes and lips contact | Dermatological beneficiation. Decongestant, anti-inflammatory, purifying, refreshing and vigorous properties. Cell regeneration stimulation and promote better blood circulation in the treated area. Strengthen the skin tension, toning and firming it. It generates skin softness [30]. |

**Table 2.** *Cont.*

| ID | Label Indications for Use | | | | | Therapeutic or Aesthetic Action |
|---|---|---|---|---|---|---|
| | Temperature | Application Dose | Action Time and Procedure | Periodic Application | Adverse Effects Caution | |
| C9 | Cold application | Not quantified ("thick layer") | Applied as a face and neck mask for 10 min. Clean with warm water, dry, moisturize with cream. Apply as a cataplasm leaving about 2–3 cm thick of clay in contact with the skin. Wrap with thin gauze and leave on for 1 h. Remove the clay, clean with warm water and dry. | - | Avoid contact with eyes and lips. Avoid sensitized skin. | Beauty mask. Effective on mixed and oily skins. Absorbent, purifying and regenerating properties It absorbs and regulates excess sebum, removes impurities, revitalizes the skin and promotes cell renewal. It generates skin softness. |
| C10 | Hot application. | - | Cataplasm and mud bath | - | - | Musculoskeletal disorders [34] |
| C11 | Cold application | Not quantified ("Make a paste with a little flower water") | Applied as a face mask. Leave for 15 min before rinsing off. May add a few drops of essential oil to the mixture. | - | - | Purifying and demineralizing, particularly suitable for gentle cleansing of oily skin. |
| C12 | Cold application | Not quantified ("Make a paste with a little flower water") | Applied as a face mask. Leave for 15 min before rinsing off. May add a few drops of essential oil to the mixture. | - | - | Highly absorbent, particularly suitable for deep cleansing of oily skin. |
| C13 | Cold application | Not quantified ("Make a paste with a little flower water") | Applied as a face mask. Leave for 15 min before rinsing off. May add a few drops of essential oil to the mixture and a teaspoon of vegetable oil to the mixture. | - | - | Soothing and remineralizer, particularly suitable for sensitive and irritated skin. |

**Table 2.** *Cont.*

| ID | Label Indications for Use | | | | | Therapeutic or Aesthetic Action |
|---|---|---|---|---|---|---|
| | **Temperature** | **Application Dose** | **Action Time and Procedure** | **Periodic Application** | **Adverse Effects Caution** | |
| C14 | Cold application | Not quantified ("Make a paste with a little flower water") | Applied as a face mask. Leave for 15 min before rinsing off. May add a few drops of essential oil to the mixture. | - | - | Highly absorbent, particularly suitable for deep cleansing of oily skin. |
| C15 | Cold application | Not quantified ("Make a paste with a little flower water") | Applied to the hair and to the hair scalp. Leave for 15 min before rinsing with a mild shampoo. May add a few drops of essential oil to the mixture. | - | - | Hair care restores shine and volume. Ghassoul allows the absorption of excess sebum and the elimination of impurities [35]. |
| C16 | Hot application | - | Applied as a cataplasm for 12–15 min. | - | - | Musculoskeletal disorders—application on the upper back |

### 2.2. Methods

2.2.1. Mineralogy and Granulometry

The samples were frozen, then freeze-dried, and gently disaggregated before testing. Grain size distribution bellow 100 μm particle size was assessed using an X-ray grain size analyzer (Sedigraph III Plus).

Mineralogical analysis of both fine and clay fractions was carried out by X-ray diffraction, using a Philips/Panalytical X'Pert-Pro MPD, Kα Cu (λ = 1,5405 Å) radiation, with $0.02°$ 2θ s$^{-1}$ steps in goniometer speed by the random-oriented powders (total sample) and oriented aggregates (<2 μm). The oriented aggregates were treated with glycerol and heat treatment at 500 °C. The identification and semi-quantification of the different mineral phases were based on measured peak areas of the basal reflections, considering the full width at half maximum, and then, weighted by empirically estimated factors or reflection powers [36,37].

2.2.2. Physicochemical Properties

The chemical composition of major and minor elements was assessed by X-Ray fluorescence using a Panalytical AX-IOS PW4400/40 X-ray fluorescence spectrometer, and it provided the data for major chemical elements: $SiO_2$, $Al_2O_3$, $Fe_2O_3$, $TiO_2$, MnO, CaO, MgO, $K_2O$, $Na_2O$, and $P_2O_5$, as well as loss on ignition (LOI) and minor chemical elements: As, Ba, Ce, Co, Cr, Cu, Ga, Mo, Ni, Pb, Rb, Sc, Sn, Th, U, V, Zn, and Zr.

The cation exchange capacity (CEC) was determined using ammonium acetate ($CH_3COOHNH_4$) as the saturation solution. This method involves the saturation of 10 g of dried sample with 200 mL of ammonium acetate for 24 h. The exchangeable cations ($Na^+$, $K^+$, $Mg^{2+}$, and $Ca^{2+}$) were carried out by ICP Mass Spectrometry (Agilent Technologies 7700Series) after collection of 100 mL of the filtered solution, under vacuum extraction, using Macherey–Nagel MN640d filter paper. The excess of ammonium acetate was rinsed with ethanol until completely cleaned by testing with Nessler reagent. Before conducting CEC analyses, 200 mL deionized water and 2 g of oxide magnesium (MgO) were added to the sample and distillate. Additionally, 100 mL of distilled solution was collected into a volumetric flask with 50 mL of boric acid solution 4% ($H_3BO_3$) and a bromocresol indicator (0.1%). CEC determination was concluded with 0.1 N chloride acid (HCl) titration.

The pH was determined by using a HANNA HI 9126 pH meter, previously calibrated with standards (Titisol standard solutions) at pH 4 and pH 7, with an accuracy of ±0.05.

The specific surface area (SSA) was estimated by BET analysis using Micromeritics Instrument Corporation—Gemini II 2370 equipment, and specific heat was found by differential scanning calorimetry (DSC) using a Dynamic Scanning Calorimeter DSC-50 Shimadzu.

Expandability was determined following the Portuguese LNEC Standard E200-1967. It was weighted with 100 g of dried sample and packed into a cylinder supported on a porous plate. The measurement was periodically made using a deflectometer, after hydration of the porous base, until constant value. The difference between initial and final measurement corresponds to the expandability/swelling result [38].

The liquid and plastic limit (LL and PL, respectively), as well as the plasticity index (PI), have been determined following an adaptation of the Portuguese standard NP143-1969 and calculated from Atterberg Limits [38,39].

It was weighted with 100 g of dried clay-based sample and was moistened with demineralized water to obtain a semisolid formulation. For the LL determination, the standard Casagrande cup was half filled with the formulation. Paste was grooved and two rotations per second of the apparatus were inflicted until the narrow cut closed. Number of rotations and weight were taken, and the moisture was dried. There were four measurements taken at decreasing water contents. LL is, by definition, the sample moisture content when the groove is closed by 25 blows. This amount was determined by the number of blows log versus water content graph projection. The PL is based on the determination of moisture content of a 3 mm diameter and 10 cm length cylinder made

with the sample when it starts to fracture. The water content is measured by the weight cylinder difference after dried-up. The difference between LL and PL is called the PI.

The abrasiveness index (AI) was determined using an Einheler AT-1000 apparatus following Neubold et al.'s (1982) procedure indications [40]. For this purpose, 50 g of dry sample was mixed with 500 mL of distilled water until complete homogenization. The mixture was put in contact with a clean and dry standard bronze wire, previously weighted, onto the Einheler apparatus, for stirring, during 174,000 revolutions ($\approx$ 96 min). For the samples that break up the net, the procedure is repeated for 87,000 revolutions or 43,500 revolutions ($\approx$ 30 min). At the end of the Einheler test, the wire is cleaned, dried, and weighted [40]. The AI is the result of the bronze wire net weight difference before and after abrasion (wear) by area value.

The relative density of the samples was determined using a pycnometer for solids, with precise measurements of weight and volume. Demineralized water was used for the working liquid.

To estimate the cooling kinetics properties of the samples, 20 g of dried sample was heated at 60 °C for 6 h. Every 30 s, a measurement of temperature was made using a dual thermometer, Lutron TM-906A, starting the measurements at 50 °C, until the sample reached 30 °C.

## 3. Results

### 3.1. Mineralogy and Granulometry

The mineralogical associations of studied samples are presented in Tables 3 and 4, considering XRD data after validation by chemical one. Quartz is always present on samples, whereas carbonates, feldspars, and iron minerals are more variable. Low quantities of dolomite are present in some samples in association with calcite; sample C7 shows Mg calcite. Samples C7 and C8 also present a low quantity of opal, which is naturally associated with the volcanic ground.

Samples are quite rich in clay minerals, except for samples C4 (7%) and C6, C7, C9, and C10 that reveal less than 35% of the clay minerals content. Samples C2, C11, C13, and C16 are almost pure (more than 80% of clay minerals). Kaolinite content is significant in sample C13 (96%). Most of the samples reveal kaolinite and illite as the main clay minerals, but occasionally, smectite is also present, except for sample C2, which reveals the highest content (100%) of smectite.

For samples C1, C2, C8, C10, C11, C13, C15, and C16, the treatment with glycerol causes the shift of smectite reflection to values around 18.5–18.9 Å, and for samples C3 C5, C6, C10, C11, and C14, they are around 15.0–17.6 Å, which indicates the presence of swelling smectites. The basal spacing exhibit by smectite, in referred samples without treatment, showed reflections around 12–15 Å, suggesting that, in these samples, smectite interlayer space was mainly occupied by divalent cations.

Grain-size data (Figure 1) indicates that samples have a high content of <2 μm fraction (>42%). Paste mud samples revealed a content of clay fraction (<2 μm) between 45% and 92% of the total fraction, whereas powder samples revealed a content between 42% and 99%. The average diameter for paste samples ranges between 0.141 μm and 3.17 μm, and for powder samples, it ranges between 0.103 and 3.29 μm.

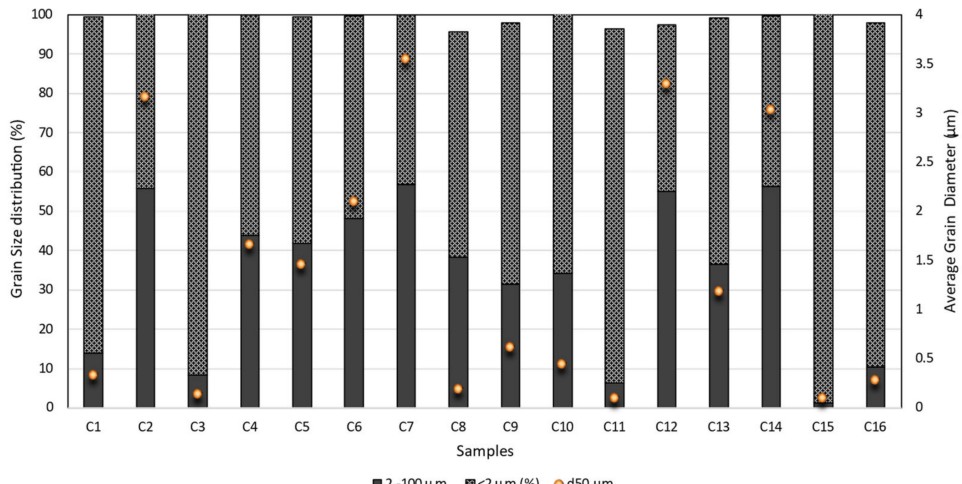

**Figure 1.** Grain size distribution and average diameter ($d_{50}$) of the clay-based samples.

**Table 3.** Mineralogical composition (wt %) of the studied samples.

|  | Qtz | Fsp | Pl | Opl | Tlc | Hmt | M-M | Cal | Dol | Gyp | Arg | Sd | Ant | Crs | Pyrt | Hlt | Zlt | Phy |
|---|---|---|---|---|---|---|---|---|---|---|---|---|---|---|---|---|---|---|
| C1 | 21 | 3 | 3 | - | - | - | - | 4 | - | - | - | - | - | - | - | - | - | 69 |
| C2 | 7 | 1.5 | 1 | - | - | - | - | trace | - | - | - | - | - | - | - | 2.5 | - | 88 |
| C3 | 11 | 6 | 23 | - | - | - | - | 6 | - | - | - | - | - | - | - | 7 | - | 47 |
| C4 | 26 | 2 | 4 | - | - | - | - | 50 | 8 | - | - | - | - | - | - | 3 | - | 7 |
| C5 | 49 | 2 | 6 | - | - | - | - | - | 1 | - | - | - | 2 | - | - | - | - | 40 |
| C6 | 13 | 2 | 4 | - | - | - | - | 37 | 6 | - | - | - | 3 | - | - | - | - | 35 |
| C7 | 6 | 5 | 3 | 11 | 5 | - | - | 1 | 9 * | - | - | - | - | - | - | - | 32 | 28 |
| C8 | 19 | - | - | 4 | - | - | 3 | - | - | - | - | - | - | - | 10 ** | - | - | 64 |
| C9 | 14 | - | - | - | - | - | - | 35 | - | - | - | 8 | 7 | - | - | 2 | - | 35 |
| C10 | 26 | 15 | 1 | - | - | - | - | 22 | - | - | - | - | - | - | 1 | - | - | 35 |
| C11 | 6 | - | - | - | - | - | - | 5 | - | - | 3 | - | - | - | - | - | - | 86 |
| C12 | 27 | 5 | 2 | - | - | - | - | - | - | 1 | - | 9 | - | 5 | - | - | - | 51 |
| C13 | 4 | trace | trace | - | - | - | - | - | trace | - | - | trace | - | - | - | - | - | 96 |
| C14 | 25 | 5 | 1 | - | - | 1 | - | 6 | - | - | - | 5 | 4 | - | - | 2 | - | 51 |
| C15 | 9 | 1 | 1 | - | - | 2 | - | - | 33 | 5 | - | trace | - | - | - | - | - | 49 |
| C16 | 15 | 6 | 4 | - | - | - | - | 4 | - | - | - | - | - | 4 | - | - | - | 77 |

Qtz, Quartz; Fsp, K-Feldspar; Pl, Na-Plagioclase; Opl, Opal; Tlc, Talc; Hmt, Hematite; M–M, Magnetite–Maghemite; Cal, Calcite; Dol, Dolomite, Gyp-Gypsum; Arg, Aragonite; Sd, Siderite; Ant, Anatase; Crs, Cristobalite; Pyrt, Pyrite; Hlt, Halite; Zlt, Zeolite and Phy, Phyllosilicates.* Mg calcite; ** Ca and Fe Sulphates.

**Table 4.** Clay fraction (wt %) of the studied samples.

|  | Phyllosilicates | | |
|---|---|---|---|
|  | Mca/Ill | Kln | Sme |
| C1 | 44 | 4 | 52 |
| C2 | - | - | 100 |
| C3 | 49 | - | 51 |
| C4 | - | 100 | trace |
| C5 | 62 | 36 | 2 |
| C6 | 48 | 39 | 13 |
| C7 | 100 | - | trace |
| C8 | 21 | 39 | 40 |
| C9 | 46 | 54 | - |
| C10 | 59 | 30 | 11 |
| C11 | 76 | - | 24 |
| C12 | 64 | 36 | - |
| C13 | - | 96 | 4 |
| C14 | 85 | 14 | trace |
| C15 | - | - | 100 |
| C16 | 18 | 2 | 80 |

Mca/Ill, Mica/Illite; Kln, Kaolinite and Sme, Smectite.

### 3.2. Physicochemical Properties

3.2.1. Chemical Composition

Geochemical data of the studied samples is provided in Table 5. There are significant variations on $SiO_2$, $Al_2O_3$, CaO, and MgO content in the samples. C16 sample shows a very low LOI, considering its phyllosilicate content; this sample was additived by the thermal spa having relevant amounts of paraffin (being defined by them as a parafang), which affects the mineralogical composition assessment. On the contrary, for sample C4, the peculiar high value of LOI is related to calcite and dolomite. MgO content in C1, C2, and C3 samples is very high; the richer, C2, has 85% of phyllosilicates (all smectite), but the other two do not actually show identified (by XRD) mineral phases as being Mg-rich, pointing to the occurrence of amorphous ones (Mg oxides/hydroxides). Sample C8, which has a phyllosilicates content dominated by smectite, shows some disagreements between mineralogical and chemical composition, which is most probably due to amorphous phases, such as on C2, but in this case, sulphates (not oxides) consider the amount of $SO_3$ (8.19%).

The principal component analysis (PCA) was used to characterize samples in what concerns the major chemical elements and LOI composition. The data matrix included the 16 samples studied and 12 variables for chemical composition. The explained variance percentage and the cumulative variance of axes are shown in Tables 6 and 7. The four axes retention was based on eigenvalues >1 criterion, and selected variables were with values > |0.5| [41]. The first four axes explained 84.61% of the total inertia. The first two axes had eigenvalues of 4.59 and 2.71, respectively, and explain 38.24% and 22.56%, correspondently, of the total inertia. Axis 1 explained $Al_2O_3$, $Fe_2O_3$, $K_2O$, $SO_3$, MnO, $TiO_2$, and $P_2O_5$ in opposition to $SiO_2$, MgO, and LOI; axis 2 explained $SiO_2$ in opposition to CaO, $P_2O_5$, and LOI; axis 3 explained MgO and $Na_2O$ in opposition to $Al_2O_3$; axis 4 explained variables already explained by the other axes.

The variable projection on PCA's first factorial plane established groups of samples according to their chemical affinity (Figure 2).

Samples C1, C2, C3, C7, C13, and C16 are positively correlated with $Na_2O$–MgO association in opposition to samples C11 and C14 that are positively correlated with CaO–$SO_3$ association.

In axis 2, samples may be distinguished by $Al_2O_3$ higher content (C5, C9, and C12) and with more $Fe_2O_3$–$TiO_2$–$P_2O_5$–MnO content (C11 and C14).

According to ordination diagram, associations between chemical composition and mineralogy signature can be established:

- sample C13's high concentration of $Al_2O_3$ reflects its high kaolinite content;
- the highest values on $K_2O$ and $Fe_2O_3$ can be associated to mica/illite contents and distinguished C5, C9, C11, C12, and C14 samples;
- samples C4 and C6 reveal higher values of CaO and LOI, which can be associated with their carbonate minerals composition (calcite and dolomite contents);
- samples C2 and C3 are differentiated by $Na_2O$ content reflecting the halite (C3) and smectite (C2) contents;
- sample C15 is distinguished by the MgO value related to its dolomite content and due to the nature of the clay, which is a magnesian smectite.

**Table 5.** Major-element content (wt %) of the studied samples.

| Sample | SiO₂ | Al₂O₃ | Fe₂O₃ | CaO | Na₂O | K₂O | SO₃ | MnO | MgO | TiO₂ | P₂O₅ | LOI * |
|--------|-------|--------|--------|-------|-------|-------|-------|-------|-------|-------|-------|--------|
| C1 | 50.73 | 9.45 | 4.23 | 1.55 | 2.15 | 2.39 | 0.08 | 0.09 | 19.25 | 0.49 | 0.19 | 8.93 |
| C2 | 49.17 | 5.14 | 1.43 | 1.01 | 4.17 | 1.07 | 1.04 | 0.03 | 22.16 | 0.25 | 0.08 | 12.47 |
| C3 | 49.20 | 7.99 | 3.39 | 1.31 | 3.36 | 1.85 | 1.11 | 0.04 | 19.43 | 0.44 | 0.05 | 10.25 |
| C4 | 17.90 | 5.34 | 3.00 | 20.95 | 0.85 | 2.25 | 0.91 | 0.05 | 7.24 | 0.51 | 0.22 | 31.71 |
| C5 | 61.75 | 19.14 | 7.11 | 0.81 | 0.60 | 2.75 | 0.10 | 0.07 | 1.25 | 1.04 | 0.29 | 4.86 |
| C6 | 35.47 | 11.11 | 4.94 | 19.48 | 0.86 | 2.51 | 0.62 | 0.08 | 4.53 | 0.51 | 0.13 | 19.37 |
| C7 | 70.83 | 12.02 | 1.29 | 1.76 | 0.16 | 3.86 | 0.01 | 0.02 | 0.70 | 0.08 | 0.02 | 9.11 |

**Table 5.** *Cont.*

| Sample | SiO$_2$ | Al$_2$O$_3$ | Fe$_2$O$_3$ | CaO | Na$_2$O | K$_2$O | SO$_3$ | MnO | MgO | TiO$_2$ | P$_2$O$_5$ | LOI * |
|---|---|---|---|---|---|---|---|---|---|---|---|---|
| C8 | 28.65 | 13.10 | 3.02 | 0.30 | 1.70 | 0.71 | 8.19 | 0.03 | 0.70 | 0.59 | 0.13 | 42.64 |
| C9 | 46.21 | 25.20 | 4.94 | 2,71 | 0.13 | 5.21 | 0.13 | 0.05 | 2.14 | 0.47 | 0.16 | 12.37 |
| C10 | 42.77 | 12.12 | 4.99 | 13.41 | 0.29 | 1.73 | 1.25 | 0.10 | 1.90 | 0.74 | 0.15 | 20.39 |
| C11 | 50.53 | 16.73 | 6.64 | 5.10 | 1.20 | 3.94 | 0.50 | 0.13 | 3.78 | 0.59 | 0.34 | 10.17 |
| C12 | 59.14 | 22.38 | 6.93 | 0.16 | 0.48 | 3.86 | 0.05 | 0.03 | 0.57 | 1.11 | 0.16 | 4.90 |
| C13 | 50.04 | 31.01 | 2.05 | 0.79 | 0.09 | 1.34 | 0.04 | <*d.l.* | 1.46 | 0.33 | 0.13 | 12.46 |
| C14 | 52.4 | 20.24 | 7.19 | 4.16 | 0.33 | 4.13 | 0.04 | 0.1 | 2.37 | 0.87 | 0.16 | 7.83 |
| C15 | 39.05 | 3.45 | 1.17 | 9.89 | 0.47 | 0.77 | 5.14 | 0.01 | 20.20 | 0.17 | 0.04 | 16.82 |
| C16 | 63.89 | 14.72 | 3.51 | 2.67 | 2.94 | 1.26 | 0.63 | 0.05 | 5.73 | 0.28 | 0.05 | 3.84 |
| Mean | 47.98 | 14.32 | 4.11 | 5.38 | 1.24 | 2.48 | 1.24 | 0.05 | 7.09 | 0.53 | 0.14 | 14.26 |
| Median | 49.62 | 12.81 | 3.87 | 2.21 | 0.73 | 2.32 | 0.56 | 0.05 | 3.07 | 0.50 | 0.14 | 11.31 |
| Min. | 17.90 | 3.45 | 1.17 | 0.16 | 0.09 | 0.71 | 0.01 | 0.00 | 0.57 | 0.08 | 0.02 | 3.84 |
| Max. | 70.83 | 31.01 | 7.19 | 20.95 | 4.17 | 5.21 | 8.19 | 0.13 | 22.16 | 1.11 | 0.34 | 42.64 |
| St. Dev. | 13.35 | 7.72 | 2.10 | 6.83 | 1.27 | 1.37 | 2.23 | 0.04 | 8.09 | 0.29 | 0.09 | 10.34 |

* Loss on ignition. *d.l.*—detection limit.

**Table 6.** 'Factor (F) loading' of the commercial samples and chemical composition, extracted by PCA.

| | F1 | F2 | F3 | F4 | F5 | F6 | F7 | F8 | F9 | F10 | F11 | F12 |
|---|---|---|---|---|---|---|---|---|---|---|---|---|
| Eigenvalue | 4.59 | 2.71 | 1.71 | 1.15 | 0.46 | 0.44 | 0.38 | 0.30 | 0.19 | 0.07 | 0.00 | 0.00 |
| Mean | 2.74 | 2.19 | 1.77 | 1.43 | 1.13 | 0.89 | 0.68 | 0.48 | 0.32 | 0.20 | 0.01 | 0.03 |
| Upper limit | 3.29 | 2.51 | 2.04 | 1.66 | 1.35 | 1.08 | 0.84 | 0.64 | 0.47 | 0.31 | 0.18 | 0.08 |
| Lower limit | 2.35 | 1.89 | 1.54 | 1.21 | 0.94 | 0.70 | 0.50 | 0.34 | 0.21 | 0.11 | 0.04 | 0.00 |
| Variance explained % | 38.24 | 22.56 | 14.24 | 9.58 | 3.80 | 3.70 | 3.18 | 2.50 | 1.57 | 0.61 | 0.02 | 0.02 |
| Cumulative Variance % | 38.24 | 60.79 | 75.03 | 84.61 | 88.41 | 92.10 | 95.28 | 97.78 | 99.36 | 99.96 | 99.98 | 100 |

**Table 7.** Studied samples' Principal Analysis Components.

| | F1 | F2 | F3 | F4 | Samples | F1 | F2 | F3 | F4 |
|---|---|---|---|---|---|---|---|---|---|
| SiO$_2$ | −0.42 | 0.84 | −0.01 | 0.03 | C1 | 0.12 | 0.67 | 1.98 | 0.25 |
| Al$_2$O$_3$ | −0.70 | 0.20 | −0.54 | 0.14 | C2 | 2.97 | 1.71 | 1.60 | 0.54 |
| Fe$_2$O$_3$ | −0.88 | −0.23 | 0.25 | 0.23 | C3 | 1.79 | 1.45 | 1.32 | 0.53 |
| CaO | 0.16 | −0.71 | 0.18 | −0.61 | C4 | 1.34 | −3.22 | 0.37 | −1.38 |
| Na$_2$O | 0.50 | 0.33 | 0.59 | 0.37 | C5 | −2.99 | 0.12 | 0.31 | 1.14 |
| K$_2$O | −0.78 | 0.08 | −0.03 | −0.31 | C6 | 0.06 | −1.83 | 0.50 | −1.41 |
| SO$_3$ | 0.60 | −0.41 | −0.30 | 0.49 | C7 | 0.40 | 2.31 | −1.59 | −1.71 |
| MnO | −0.56 | −0.37 | 0.61 | −0.03 | C8 | 2.47 | −2.38 | −1.91 | 2.61 |
| MgO | 0.64 | 0.25 | 0.60 | 0.02 | C9 | −1.87 | 0.27 | −1.16 | −0.72 |
| TiO$_2$ | −0.72 | −0.33 | 0.09 | 0.42 | C10 | −0.53 | −1.79 | 0.25 | −0.40 |
| P$_2$O$_5$ | −0.66 | −0.47 | 0.27 | 0.19 | C11 | −2.62 | −0.90 | 1.45 | 0.20 |
| LOI | 0.49 | −0.77 | −0.27 | 0.15 | C12 | −2.76 | 0.75 | −0.78 | 0.81 |
| | | | | | C13 | −0.02 | 1.16 | −2.50 | −0.16 |
| | | | | | C14 | −2.77 | −0.18 | 0.26 | 0.03 |
| | | | | | C15 | 3.65 | −0.22 | −0.44 | −0.50 |
| | | | | | C16 | 0.77 | 2.07 | 0.34 | 0.15 |

The highest concentrations of potential toxic elements were present on the listed samples: C11 (As-44.7 ppm); C14 (Ba-690 ppm); C12 (Cr-230 ppm); C6 (Ni-83.1 ppm); C13 (Pb-61.3 ppm); C5 (Zn-130 ppm) (Table 8).

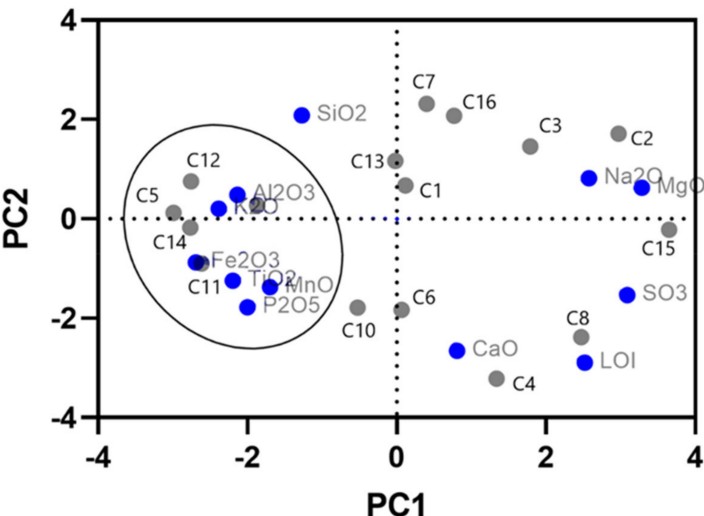

**Figure 2.** Studied samples' Principal Analysis Components projection (PC1/PC2).

**Table 8.** Trace element content (ppm) of the studied samples and the acceptable limits for heavy metals in topic products [42–44].

| Sample | As [2] | Ba [1] | Cd [2] | Ce [4] | Co [2] | Cr [2] | Cu [3] | Ga [4] | Mo [4] | Ni [2] | Pb [2] | Rb [4] | Sc [4] | Sn [4] | Th [2] | U [4] | V [4] | Zn [5] | Zr [2] |
|---|---|---|---|---|---|---|---|---|---|---|---|---|---|---|---|---|---|---|---|
| Acc. L. | 3 | 1300 | 3 | | 5 | 1100 | 130 | | 18 | 60 | 10 | | | | | | | | |
| C1 | 16.4 | 150 | <*d.l.* | 27.9 | 7.6 | 31.6 | 20.0 | 13.1 | 1.2 | 8.3 | 20.1 | 140 | 5.5 | 8.9 | 9.1 | 7.8 | 58.8 | 59.7 | 83.4 |
| C2 | 18.5 | 620 | <*d.l.* | 23.9 | <*d.l.* | 20.6 | 15.9 | 6.1 | <*d.l.* | 4.3 | 16.2 | 53.9 | <*d.l.* | 6.3 | 7.7 | 3.9 | 27.4 | 23.3 | <*d.l.* |
| C3 | 18.8 | 190 | <*d.l.* | 37.3 | 5.5 | 28.2 | 18.9 | 11.5 | 1.1 | 8.2 | 10.5 | 95.1 | 3.9 | 8.5 | 9.0 | 6.2 | 41.4 | 56.2 | 84.9 |
| C4 | <*d.l.* | 170 | <*d.l.* | 4.4 | 8.8 | 90.5 | 18.7 | 7.2 | 3.4 | 26.3 | <*d.l.* | 32.0 | 14.9 | <*d.l.* | 7.5 | 4.4 | <*d.l.* | 50.8 | 260 |
| C5 | 23.5 | 380 | <*d.l.* | 97.2 | 16.6 | 99.0 | 21.5 | 20.6 | 1.3 | 38.7 | 33.4 | 150 | 12.3 | 6 | 15.9 | 4.7 | 150 | 130 | <*d.l.* |
| C6 | 5.0 | 330 | <*d.l.* | 56.1 | 14.6 | 170 | 29.1 | 11.8 | 2.5 | 83.1 | 21.6 | 530 | 17.7 | <*d.l.* | 10.4 | 5.0 | 93.7 | 70.0 | 71.2 |
| C7 | <*d.l.* | 77.1 | <*d.l.* | 4.8 | <*d.l.* | <*d.l.* | 4.0 | 15.2 | <*d.l.* | <*d.l.* | 21.0 | 230 | 4.9 | 7.4 | 21.7 | 4.8 | <*d.l.* | 37.8 | 130 |
| C8 | <*d.l.* | 260 | <*d.l.* | 60.0 | 5.3 | 42.1 | 14.7 | 10.7 | 4.9 | 15.5 | 19.4 | 9.1 | 8.8 | 4 | 10.6 | 2.3 | 64.0 | 28.3 | <*d.l.* |
| C9 | 17.7 | 260 | <*d.l.* | 43.9 | 7.5 | 46.5 | 28.6 | 31.5 | 1.1 | 21.8 | 26.7 | 400 | 6.4 | 18 | 10.6 | 4.4 | 55.70 | 82.8 | 47.4 |
| C10 | 10.7 | 500 | <*d.l.* | 66.4 | 13.2 | 78.5 | 49.1 | 13.3 | 0.9 | 31.9 | 37.7 | 87.2 | 12.9 | <*d.l.* | 8.9 | 2.8 | 71.30 | 100 | 180 |
| C11 | 44.7 | 340 | 4.1 | 75.8 | 9.2 | 180 | 27.9 | 21.7 | 1.8 | 26.7 | 44.5 | 220 | 11.3 | 8.9 | 20.1 | 7.9 | 85.4 | 94.2 | 51.3 |
| C12 | 9.6 | 350 | <*d.l.* | 120 | 14.0 | 230 | 39.6 | 21.9 | 1.3 | 61.4 | 41.9 | 170 | 14.7 | 6.4 | 16.3 | 4.2 | 130 | 99.6 | 310 |
| C13 | <*d.l.* | 340 | <*d.l.* | 110 | <*d.l.* | 23.6 | <*d.l.* | 32.8 | 0.9 | 2.3 | 21.6 | 83.1 | <*d.l.* | 14.9 | 14.6 | 5.4 | 27.1 | 9.5 | 72.2 |
| C14 | 21.6 | 690 | <*d.l.* | 85.2 | 14.8 | 84.0 | 30.6 | 19.9 | 1.7 | 32.3 | 28.9 | 180 | 13.4 | 7.3 | 15.5 | 3.2 | 120 | 92.3 | 250 |
| C15 | 9.3 | 130 | <*d.l.* | 16.2 | 14.1 | 25.7 | 15.4 | 5.7 | 2.5 | 7.1 | <*d.l.* | 35.6 | 5.7 | <*d.l.* | 48.3 | 32.0 | 260 | 17.5 | <*d.l.* |
| C16 | 5.1 | 260 | <*d.l.* | <*d.l.* | <*d.l.* | 36.7 | 12.8 | 13.8 | 1.6 | 16.0 | 23.6 | 60.6 | 5.8 | 4.5 | 13.7 | 3.5 | 23.8 | 46.3 | 190 |
| Mean | 16.7 | 315 | - | 55.3 | 10.7 | 79.1 | 23.7 | 16.1 | 1.9 | 25.6 | 29.1 | 154.8 | 9.9 | 8.4 | 15.0 | 6.4 | 86.3 | 62.4 | 115.4 |
| Min. | 5.0 | 77.1 | - | 4.4 | 5.3 | 20.6 | 4.0 | 5.7 | 0.9 | 2.3 | 10.5 | 9.1 | 3.9 | 3.9 | 7.5 | 2.3 | 23.8 | 9.5 | 0.0 |
| Max. | 44.7 | 690 | 4.1 | 120 | 16.6 | 230 | 49.1 | 32.8 | 4.9 | 83.1 | 61.3 | 530 | 17.7 | 18.1 | 48.3 | 32.0 | 260 | 130 | 310 |
| St. Dev. | 10.8 | 171 | - | 37.1 | 4.1 | 65.6 | 11.5 | 8.1 | 1.1 | 22.4 | 13.5 | 140.2 | 4.5 | 4.1 | 9.9 | 7.0 | 63.6 | 34.9 | 100.5 |

*d.l.*—detection limit; Acc. L.—Acceptable Limit. [1] Not listed as element in EC 1223/2009. [2] Not allowed in EC 1223/2009. [3] Allowed in EC 1223/2009. [4] Not listed in EC 1223/2009. [5] Allowed under specific restriction conditions in EC 1223/2009.

### 3.2.2. Cation Exchange Capacity

Exchangeable cations (EC) and CEC (cation exchange capacity) are variable (Table 9); CEC value is, in some samples, relatively unrelated to the semi-quantitative mineralogical composition, especially with the smectite content. CEC is assessed on fine fraction, and mineralogical composition (Table 3) is related to the whole sample.

The highest CEC value was achieved for C2 sample (45 meq/100 g), which was richer in phyllosilicates (clay faction 100% smectite), and the lowest was for C4 and C8 samples (1 meq/100 g). The values of CEC higher than 40 meq/100 g were registered on samples C2 and C16 with the main exchangeable cation $Na^+$. The samples with illite predominance present a CEC value between 4–34 meq/100 g.

Except for sample C7, which presents $K^+$ as the main exchangeable cation, and C1, C2, C3, and C16, which present $Na^+$, the remaining samples reveal $Ca^{2+}$ as the main exchangeable cation (values between 48 mg/L and 1225 mg/L). The samples C1, C7, and C10 have an approximate value for CEC despite the difference of the exchangeable cations.

**Table 9.** Cation exchange capacity and individual main exchangeable cations.

| Sample | C1 | C2 | C3 | C4 | C5 | C6 | C7 | C8 | C9 | C10 | C11 | C12 | C13 | C14 | C15 | C16 |
|---|---|---|---|---|---|---|---|---|---|---|---|---|---|---|---|---|
| $Na^+$ | 379 | 591 | 561 | 362 | 17 | 62 | 18 | 163 | 4 | 17 | 350 | 3 | 15 | 10 | 179 | 1536 |
| $Mg^{2+}$ | 83 | 124 | 117 | 357 | 7 | 38 | 20 | 9 | 12 | 77 | 104 | 9 | 104 | 51 | 595 | 58 |
| $K^+$ | 29 | 12 | 12 | 133 | 2 | 9 | 271 | 5 | 25 | 37 | 127 | 6 | 23 | 18 | 56 | 80 |
| $Ca^{2+}$ | 252 | 182 | 217 | 825 | 54 | 719 | 253 | 48 | 452 | 985 | 1121 | 67 | 495 | 850 | 1225 | 623 |
| Σ Cat | 743 | 908 | 907 | 1676 | 79 | 828 | 562 | 226 | 492 | 1115 | 1702 | 85 | 637 | 928 | 2055 | 2297 |
| C.E.C. | 34 | 45 | 27 | 1 | 11 | 5 | 33 | 1 | 4 | 31 | 14 | 3 | 13 | 7 | 14 | 43 |
| E.C. | Na | Na | Na | Ca | Ca | Ca | K | Ca | Ca | Ca | Ca | Ca | Ca | Ca | Ca | Na |

C.E.C. (meq/100 g) and E.C. (mg/L).

### 3.2.3. pH

In Table 10, the studied samples present pH values ranging from 3.96 and 10.30. The C8 sample is volcanic, and its mineralogy justifies the lower pH, while the other samples have an average pH of 8.04 and a standard deviation of 1.05.

**Table 10.** pH values of the studied samples.

| Sample | C1 | C2 | C3 | C4 | C5 | C6 | C7 | C8 | C9 | C10 | C11 | C12 | C13 | C14 | C15 | C16 |
|---|---|---|---|---|---|---|---|---|---|---|---|---|---|---|---|---|
| pH | 10.30 | 9.55 | 8.64 | 8.00 | 7.60 | 6.75 | 7.00 | 3.96 | 6.21 | 7.45 | 7.69 | 7.70 | 8.18 | 8.34 | 8.36 | 8.83 |

### 3.2.4. Specific Surface Area, Expandability, Abrasiveness Index, and Relative Density

The technological analyses' results for specific surface area, expandability, abrasiveness, and relative density are presented in Table 11.

**Table 11.** Sample technological analysis.

| Sample | C1 | C2 | C3 | C4 | C5 | C6 | C7 | C8 | C9 | C10 | C11 | C12 | C13 | C14 | C15 | C16 |
|---|---|---|---|---|---|---|---|---|---|---|---|---|---|---|---|---|
| LL | 235 | 111 | - | 32 | 51 | 52 | 52 | 74 | 56 | 62 | 131 | 46 | 75 | 44 | 129 | - |
| PL | 49 | 36 | - | 17 | 18 | 22 | 29 | 29 | 21 | 25 | 34 | 21 | 26 | 18 | 37 | - |
| PI | 186 | 75 | - | 15 | 33 | 30 | 23 | 45 | 35 | 37 | 97 | 25 | 49 | 26 | 92 | - |
| Sw | 93 | 81 | 91 | 13 | 7 | 10 | 4 | - | 9 | 32 | 40 | 6 | 24 | 11 | 86 | 84 |
| SSA | 81 | 25 | 39 | 5 | 14 | 6 | 22 | 1 | 39 | 40 | 70 | 8 | 11 | 11 | 46 | 36 |
| Density | 1.92 | 1.90 | 1.59 | 2.35 | 2.83 | 2.56 | 2.10 | 1.74 | 2.25 | 2.59 | 2.01 | 2.52 | 2.29 | 2.58 | 2.68 | 2.11 |
| SpHeat | 638 | 558 | 539 | 888 | 104 | 165 | 343 | 90 | 166 | 739 | 448 | 60 | 399 | 216 | 740 | - |

L.L.—Liquid Limit (%); P.L.—Plastic Limit (%); P.I.—Plasticity Index (wt %); Sw,—Swelling (%); S.S.A—Specific Surface Area ($m^2$/g); Density ($\rho$s); SpHeat—Specific Heat (j/kg.K).

The plasticity properties of the samples showed a high range of value variation (Figure 3). The liquid limit (LL) values are variable between 32% (sample C4) and 235% (sample C1). Therefore, according to Bain (1971) [45] classification, samples can be distinguished in high plasticity clays due to the LL results above 50%, except for C4, C14, and C12 samples, which are classified as low plasticity clays because of their LL result under 50%.

The plastic limit (PL) results reveal values between 15% (sample C4) and 186% (sample C1). Classification according to Jenkins [45] assumes that soils are considered with high plasticity when P.L. is above 15%; thus, all samples studied are evaluated as highly plastic. Samples revealed high plasticity according to Atterberg limits because they present high plasticity when PI is above 15%. Sample C4 showed the lowest plasticity index (15%), and C1 showed the highest plasticity index (186%) for a median value of 55% for all samples. It was not possible to test samples C3 and C16 regarding their high LL. Although it was possible to measure samples C1 and C2, samples C11 and C15 also reveal higher LL results and, therefore, better plasticity behavior among the studied samples. These samples should exhibit the higher water retention capacity and, consequently, be the most appropriate to form pastes with good consistency for topic application.

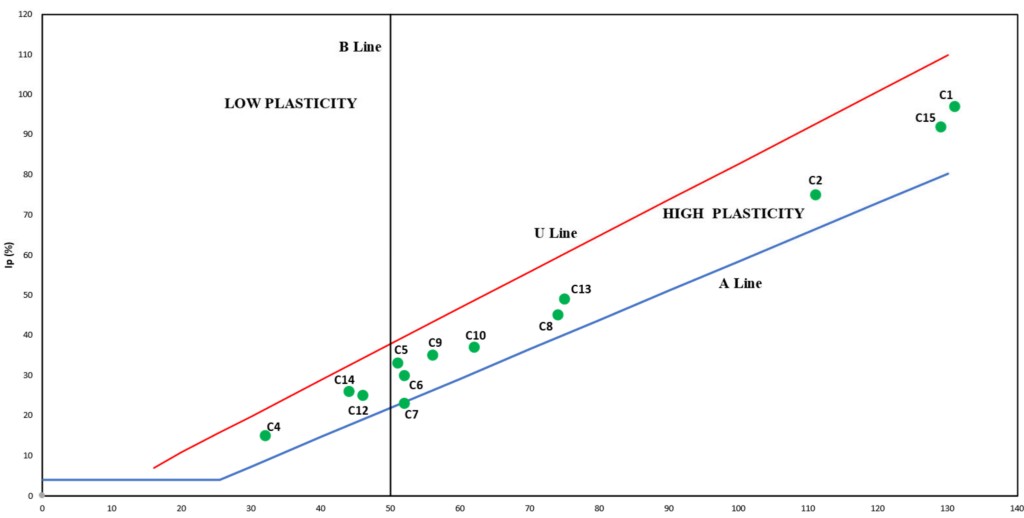

**Figure 3.** Projection of plasticity indices and liquid limits in the Casagrande Chart. (U Line, limit above which raw materials do not turn out to be plastic: LL ≥ PL; A Line, frontier between clays with or without organic colloids and B Line, LL = 50%, frontier between low (LL < 50%) and high (LL > 50%) plasticity clays).

The specific surface area (SSA) of minerals is an important parameter to quantify the clay minerals' dissolution and adsorptive interactions. The grain size distribution and clay mineralogy have important roles on SSA properties. All samples have a predominant content of <2 μm fraction and different mineralogy, showing values that are sometimes far away from what is expected from the mineralogical composition and literature [46]; studied samples are not pure clays, containing (at least some of them) amorphous components (oxides/hydroxides and/or sulphates) and additives (even organics), disturbing the theoretical relationships between mineralogical composition and technological parameters.

The lowest SSA, <11 $m^2/g$, is found in samples C4, C6, C8, C12, C13, and C14, and the highest values are in samples C1 (81 $m^2/g$) and C11 (70 $m^2/g$). The remaining samples present SSA values between 22 $m^2/g$ and 46 $m^2/g$.

The density samples present values around $2\rho$s and $3\rho$s. The specific heat enhances the samples C5, C6, C8, and C12 for the lower heat capacity ($\leq$104 j/kg.K).

The abrasiveness index indicates the ability of a sample to cause abrasion and could differentiate the use purpose of a product. Results reveal that abrasiveness varies between 296 (sample C15) and 9 (sample C16) at 174,000 rpm, and the median value is 148 $g/m^2$. Samples C4, C5, C8, C10, C12, C14, and C15 have the AI above 100 $g/m^2$ (Table 12), detaching C5, C6, C12, and C14 as the most abrasive, followed by C4 and C10.

### 3.2.5. Cooling Kinetics

The cooling rate ranged between 8 min (sample C1 and C3) and 56 min (sample C15, with 100% on clay fraction) (Figure 4). The higher cooling rates were achieved by C10, C11, C15, and C16 (>45 min). Samples C1, C2, and C3 achieve quick cooling. All the other samples, except C7 for oral intake (geophagy), which are intended for aesthetic or thermal mud formulation, cooled from 60 °C to 30 °C within 25–56 min, which is favorable to the established therapeutic procedures.

The measurement of the dry clay may be an indicator of how it will be the thermal kinetics when prepared for pelotherapy use.

**Table 12.** Einlehner abrasivity of the studied samples.

| Sample | Abrasion (mg) | | | Abrasivity Index (g/m²) |
|---|---|---|---|---|
| | **174,000 \*** | **87,000 \*** | **43,500 \*** | |
| C1 | 13 | - | - | 41 |
| C2 | 29 | - | - | 95 |
| C3 | 9 | - | - | 28 |
| C4 | - | 102 | - | 335 |
| C5 | - | - | 55 | 180 |
| C6 | - | - | 22 | 72 |
| C7 | 77 | - | - | 252 |
| C8 | - | - | - | - |
| C9 | 16 | - | - | 53 |
| C10 | - | 89 | - | 292 |
| C11 | 21 | - | - | 69 |
| C12 | - | - | 75 | 247 |
| C13 | 7 | - | - | 23 |
| C14 | - | - | 71 | 231 |
| C15 | 90 | - | - | 296 |
| C16 | 3 | - | - | 9 |

\* Revolutions.

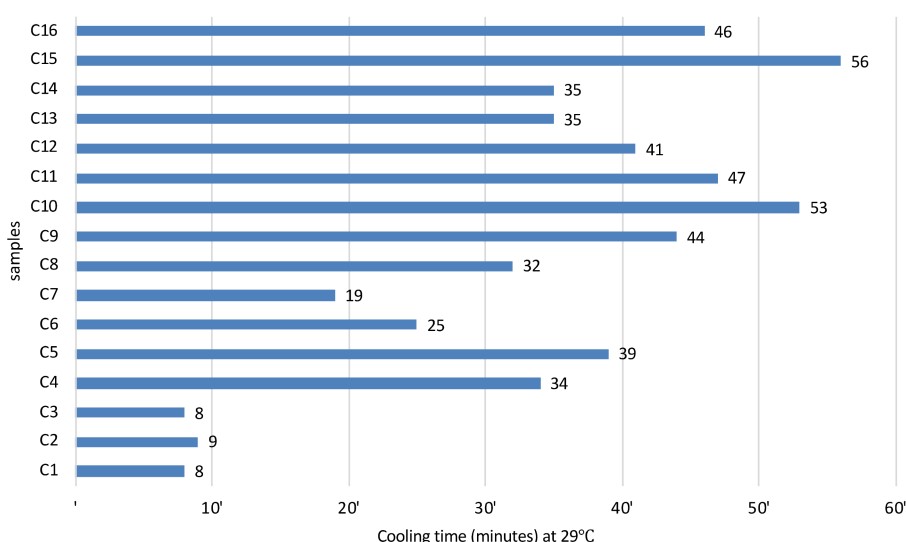

**Figure 4.** Cooling kinetics of the studied samples, from 60 °C to 30 °C.

## 4. Discussion

Personal care products (PCP) that contain substances of natural origin are considered, by the natural and organic cosmetics consumers, safe alternatives to synthetic ingredients. The main orientations in clay-based products and peloids cover, beside the mineralogical and chemical composition characterization of the raw materials for a basic formulation, the interaction mechanism between the product and the skin to aid consumers' therapeutic or aesthetic expectations, ensuring toxicological safety and low risk to the health.

The toxicological effects of clay-based hazardous components exceeding the recommended level by the international regulations may be difficult to avoid, during the raw material selection and manufacturing process, because of the high variability on toxic minerals and/or trace chemical elements in the geological materials. Considering factors such as how humans are exposed to heavy metals, how often, and in what amounts, surveys should be conducted to evaluate the extent of the toxicity of elements such as arsenic, cadmium, chromium, cobalt, lead, mercury, and nickel.

The safety evaluation of cosmetics is settled out in the European Regulation on cosmetic products (EC No. 1223/2009), as amended, with a list of more than 1300 substances

and group substances that cannot be included in the composition of cosmetic products (Annex II). A list substance, which is a cosmetic or PCP, may be contained only under the restrictions and conditions laid down (Annex III).

The EC No.1223/2009 also contains lists of colorings (Annex IV), preservatives (Annex VI), and UV filters (Annex VII) permitted in cosmetic products. Germany reduced, even more, those limits for heavy metals in cosmetics and PCP, according to the typology: cosmetic products in general and toothpaste (Lead 2.0 and 0.5 mg/kg; Arsenic 0.5 and 2.5 mg/kg, respectively; Mercury 0.1 mg/kg; Cadmium 0.1 mg/kg; Antimony 0.5 mg/kg and Nickel's limit remained unchangeable at 10 mg/kg) [44].

The United States FDA (Food and Drug Administration) also limits the presence of metals in cosmetics, as well as prohibits and restricts ingredients, by conducting regular surveys of cosmetics on the market [47].

It is urgent to clarify the classification of natural mineralogical products or mineralogical PCP ingredients and provide their declared information (CAS number) for the safe use of chemicals and replacement of substances that give cause for concern, with the supervision of ECHA—European Chemicals Agency, and they need to be registered according to REACH regulation. Cosmetic and personal care products' chemical safety is an important issue with transversal concerns, documented in several studies [48–54].

As cosmetics and medicinal products are intended to be used topically or orally, technological features must be evaluated and adequate for the intended use. The technological characterization of materials is also imperative on clay-based products, mainly when used directly on the skin, such as peloids, which have no recommendation values or quality criteria established.

Taking into consideration that cosmetics and peloids have the same exposure route, it is reliable to assume that the technologic characterization of clay-based commercial products for aesthetic purposes may contribute to the establishment of normative values and criteria for peloids. However, when a clay-based product or peloid also meets the definition of the medicinal product, it will oblige the required compliance for the intended use.

For this study, the 13 clay-based products were acquired with the assumption that these were well accepted by the user and, therefore, presented pleasant and adequate characteristics for dermatological application. The same assumption is applied for the thermal peloids: they are applied under therapeutic personnel supervision in thermal centers and welcomed by their users.

Samples reveal a polymineralic association with the presence of variable quantities of quartz, calcite, and feldspars, whereas clay minerals are predominant and common to other peloid compositions [16,55,56].

The mineralogical composition of samples reflects the chemical signature, which was evaluated by PCA analysis and could be affecting some technological features exhibited by samples, such as plasticity, abrasivity index, and cation exchange capacity. Nevertheless, some technological property values seem to be inconsistent with mineralogical analysis results; our samples are commercial products, not pure clays, having (at least some of them) amorphous components (oxides/hydroxides and/or sulphates) and additives (even organics), disturbing the theoretical relationships between mineralogical composition and technological parameters.

Some elements, such as Ca, Mg, K, and Na, are considered essential elements for human health [56,57]. Recent studies proved the chemical elements mobilization between thermal muds and artificial sweat—or even human skin—with prominence to the significant supply of Na and Ca [11,18]. Samples studied show $Ca^{2+}$ as the main exchangeable cation, which can be a useful feature concerning human health. The CEC results suggest that cation exchange may be influenced by the particle median size, as well as clay minerals content. This fact can explain the CEC results, which are relatively low when compared to some reference values (kaolinite: <15–20 meq/100 g; illite and chlorite: 10–40 meq/100 g; smectite: 100–200 meq/100 g) [45,58].

Skin pH plays an important role in its protection mechanism. The skin's natural pH is slightly acidic, and its values range from 4.1 to 5.8, which is important to prevent the development of bacteria and the maintenance of natural flora. Inadequate skin pH may contribute to various dermatological infirmities [59]. The pH assessment of peloids is an important parameter not only to predict the risk of developing some side effects of the skin but also to stabilize its natural balance. Alkaline clay-based samples are out of the skin's natural pH, which should be paid attention to as quality and safety control.

Plasticity behavior may be affected by several physico-chemical parameters such as mineralogy, particle size, aggregation/disaggregation situation, and CEC. The studied samples reveal plasticity index values above the reference for kaolinite (26–37) but, as expected, lower than the smectite reference value (101–251) [45]. Nevertheless, all samples were considered highly plastic, which makes them suitable during their manipulation and application.

Abrasiveness and cooling kinetics can characterize the handling, pleasant sensation, and drying time of the products into the skin. Samples with higher AI may be more adequate for the exfoliating purpose and cleansing, which is ideal for establishing skin balance.

The association between abrasivity index (AI) and mineralogical signature of samples can be made. Sample C13, on PCA analysis, reflected kaolinite content and revealed lower AI (13 g/m2) when compared to groups formed by the association with carbonate minerals (samples C4, C6, and C8) or illite and iron minerals (samples C5, C9, C11, C12, and C14).

The highest abrasiveness behavior occurs on samples with higher content on detrital minerals, such as quartz, feldspars, and iron minerals (samples C4, C5, C6, C10, C12, and C14). The clay-based samples, C4, C5, C6, and C10, are semi-solid formulations (paste), with mineral medicinal water as an ingredient, and C12 and C14 are natural clays in form of powder.

The lowest abrasiveness behavior of samples C1, C2, C3, C9, and C16 may be explained by the fact that these samples are the result of a blend of natural origin ingredients and other artificial performer additives, defined by Gomes et al. (2013) as Paramud or Parapeloid [2].

Cooling kinetics characterize the sample's ability to retain or release heat. This feature is more relevant in what concerns thermal mud application because, in most cases, the intent of use is to sustain the product temperature above corporal temperature for about 15 to 20 min. The measurement of the cooling rate was made on dry samples, which means that an improvement of cooling rate is expected with the moisturizing of materials [13,16,60,61].

The concentration of trace elements (As, Ba, Cd, Ce, Co, Cr, Cu, Ga, Mo, Ni, Pb, Rb, Sc, Sn, Th, U, V, Zn, and Zr) in each sample was studied to evaluate the health risk through exposure to different elements. The presence of hazardous elements must be previously determined to evaluate the risk of dermal exposure to toxic elements. The clay-based products contain median values of 17 ppm As, 315 ppm Ba, 79 ppm Cr, 11 ppm Co, 29 ppm Pb, 26 ppm Ni, and 62 ppm Zn. Only sample C11 presented 4.1 ppm of Cd. It was not detected with As, Co, Cr, Mo, Ni, and V at sample C7 for ingestion intake. Arsenic is mostly found in a free state; it is formed as sulfur, oxygen, and iron compounds. Therefore, it must be free to cause a skin problem.

Table 13 presents the technological properties of the samples studied and their framework with their qualitative requirements for skin and clay-based product interaction previously debated. It also introduced the relevance in comparative terms with a clay-based product, for ingestion intake, certified as a medical device.

**Table 13.** Studied samples' qualitative requirements establishment by technological, thermal, and rheological properties.

| Samples | | Water Retention Capacity | | Handling Consistency Adhesiveness | | | Pleasant Sensation | | Heat Capacity | | Therapeutic Action | | Toxic or Allergenic Risk (Trace Elements) | | | | | | Skin Balance Risk |
|---|---|---|---|---|---|---|---|---|---|---|---|---|---|---|---|---|---|---|---|
| | | Liquid Limit % | Expandability % | Clay Minerals | Clay Average Diameter $d_{50}$ μm | Plasticity Index Wt% | Density $\rho s$ | Abrasivity Index g/m² | Cooling Kinetics Minutes | Specific Heat j/kg.K | Exchangeable Cation | Specific Surface Area m²/g | As | Ba | Cr | Ni | Pb | Zn | pH |
| Peloid | C5 | 51 | 7 | Ill | 1.5 | 33 | 3 | 180 | 39 | 104 | Ca | 14 | x | x | x | x | x | x | 7.60 |
| | C10 | 62 | 32 | Ill | 0.4 | 37 | 3 | 292 | 53 | 739 | Ca | 40 | x | x | x | x | x | x | 7.45 |
| | C16 | - | 84 | Sme | 0.3 | - | 2 | 9 | 46 | - | Na | 36 | x | x | x | x | x | x | 8.83 |
| Clay-based product (paste) | C1 | 235 | 93 | Sme | 0.3 | 186 | 2 | 41 | 8 | 638 | Na | 81 | x | x | x | x | x | x | 10.30 |
| | C2 | 111 | 81 | Sme | 3.2 | 75 | 2 | 95 | 9 | 558 | Na | 25 | x | x | x | x | x | x | 9.55 |
| | C3 | - | 91 | Sme | 0.1 | - | 2 | 28 | 8 | 539 | Na | 39 | x | x | x | x | x | x | 8.64 |
| | C4 | 32 | 13 | Kln | 1.7 | 14 | 2 | 335 | 34 | 888 | Ca | 5 | | x | x | x | | x | 8.00 |
| | C6 | 52 | 10 | Ill | 2.1 | 30 | 3 | 72 | 25 | 165 | Ca | 6 | x | x | x | x | x | x | 6.75 |
| | C8 | 74 | - | Sme | 0.2 | 45 | 2 | - | 32 | 90 | Ca | 1 | | x | x | x | x | x | 3.96 |
| | C9 | 56 | 9 | Kln | 0.6 | 35 | 2 | 53 | 44 | 166 | Ca | 39 | x | x | x | x | x | x | 6.21 |
| Clay-based product (powder) | C11 | 131 | 40 | Ill | 0.1 | 97 | 2 | 69 | 47 | 448 | Ca | 70 | x | x | x | x | x | x | 7.69 |
| | C12 | 46 | 6 | Ill | 3.3 | 25 | 3 | 247 | 41 | 60 | Ca | 8 | x | x | x | x | x | x | 7.70 |
| | C13 | 75 | 24 | Kln | 1.2 | 49 | 2 | 23 | 35 | 399 | Ca | 11 | | x | x | x | x | x | 8.18 |
| | C14 | 44 | 11 | Ill | 3.0 | 26 | 3 | 231 | 35 | 216 | Ca | 11 | x | x | x | x | x | x | 8.34 |
| | C15 | 129 | 86 | Sme | 0.1 | 92 | 3 | 296 | 56 | 740 | Ca | 46 | x | x | x | x | | | 8.36 |
| Ingestion Intake | C7 | 52 | 4 | Ill | 3.5 | 23 | 2 | 252 | 19 | 343 | K | 22 | | x | | | x | x | 7.00 |

## 5. Conclusions

Different countries regulate these clay-based products under different frameworks. The absence of premarket approval, mainly for clay-based products that are intended for a transdermal therapeutic purpose, such as treating or preventing infirmities, makes it difficult for consumers to determine the safety and effective clinical compliance. The importance of labeling uniformity and product safety information, ensuring the use of warning statements, is also an issue of quality compliance that is missing.

The mineralogical composition of the 16 studied samples reveals a polymineralic association with the presence of variable quantities of quartz, calcite, and feldspars, whereas clay minerals are not predominant and characterized by the presence of a clay-based fraction content, composed mainly by illite, smectite, and kaolinite in variable amounts and several mineral associations. The mineralogical composition of the studied samples is not always in accordance with the label information, which is the case for C1 and C3 samples.

The studied samples have safety concerns about specific limits of As, Ba, Cd, Cr, Co, Pb, Ni, and Zn, which are above the regulated avoidable limits. Additionally, the pH of the samples is out of range of the skin's natural pH.

It was important to compare the results of a clay-based product for ingestion intake, which were certified and safe monitored (pharmacovigilance). The postmarketing surveillance databases of these clay-based medical devices, which monitor the adverse reactions and product effectiveness, can be a way to previously refine the safety of a topical clay-based product or peloid.

**Author Contributions:** Conceptualization, C.M.B. and F.R.; methodology, C.M.B. and F.R.; resources, C.M.B. and F.R.; investigation, C.M.B. and F.R.; writing—original draft preparation, C.M.B.; writing—review and editing, C.M.B. and F.R.; supervision, F.R.; project administration, C.M.B.; funding acquisition, C.M.B. All authors have read and agreed to the published version of the manuscript.

**Funding:** This research was funded by FCT—Fundação para a Ciência e a Tecnologica and Exatronic, Lda., grant number SFRH/BDE/11062/2015 and also supported by GeoBioTec Research Center (UIDB/04035/2020), funded by FCT, FEDER funds through the Operational Program Competitiveness Factors—COMPETE.

**Institutional Review Board Statement:** Not applicable.

**Informed Consent Statement:** Not applicable.

**Data Availability Statement:** Not applicable.

**Acknowledgments:** We are grateful to Balneario El Raposo, Termas de Monchique and Régie Municipale des Boues de Dax for providing samples for this study, of great clinical relevance in the treatment and prevention of musculoskeletal disorders.

**Conflicts of Interest:** The authors declare no conflict of interest.

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
