# Peer review of "Assessment of Some Clay-Based Products Available on Market and Designed for Topical Use"

_geosciences, doi:10.3390/geosciences12120453_

Round 1

Reviewer 1 Report

I have written my suggestions in detail at many points on the text. This study is in the form of a case study, it does not provide a new knowledge or method application to the situation known in the field. Researchers wrote some of the interpretation but the main deficiency of this study is to "clarify an unknown subject, suggest a method for the use of the samples, etc. In short, the main problem that needs to be solved scientifically in research is insufficient, it is repetition of what is known and data evaluation. By making these analyzes, which important problem will be solved, and contribution will be made to the evaluation of the materials must be explained. There is no standard recommendation in the evaluation of peloid, for example, it does not include an evaluation in terms of mineralogical, chemical composition and physical properties.

The word “clayey” is in the title of the work and in many parts of the text. However, in most of the samples examined, the clay content is below 20%. Also, not all phyllosilicate minerals are clay minerals. As such, there are inconsistencies with the research results given in the literature and within itself. In particular, the main element oxide contents do not seriously match with the calculated mineralogical composition content. Similarly, the mineralogical composition also affects many physical properties (eg. consistency limits, CEC, SSA, abrasivity, etc) of the material under investigation. The analysis results of these properties are also inconsistent with the mineral contents. 

Author Response

We thank the helpful comments. We have carefully revised the manuscript according to the reviewer’s insightful comments and provide point-by-point responses as follows. Please see the attached pdf 

Sincerely

Reviewer 2 Report

Authors must review the manuscript and correct it, taking into account the following remarks:

P2: incomplete sentence to be corrected:

The pharmaceutical and cosmetic industry uses clays in their formulations and? is subject to prior control before being used [10,11]

P9: make AI in place of IA in this sentence “The IA is the result of the bronze wire net weight difference before and after abrasion (wear) by area value."

P10: the C15 sample is the Ghassoul of Morocco, it is necessary to check, it is not formed of Kaolinite but of magnesian smectite of the stevensite type.

It is necessary to correct in all the tables and the different interpretations.

P11: "sample C15 is distinguished by the MgO value related to its dolomite content”

The high rate of MgO is also due to the nature of the clay which is a magnesian smectite.

P13: The values ​​of CEC higher than 40 meq/100g were registered on samples C2 and C3, with the main exchangeable cation Na+.

You have to put C2 and C16

It is strange to see the CEC of the C3 sample relatively high, yet the clayey part is not very important (38%) and it is illite.

P15: To check the plasticity of the sample C1, which is not found in figure 3

Similarly, the terms Low plasticity and High plasticity must be moved a little to the left, which are separated by line B

P15: You must write “The higher cooling rates were achieved by C10, C11, C15 and C16 (> 45 mn) instead of “The higher cooling rates were achieved by C10, C15 and C16 (> 45℃)”.

There are several numbering errors that must be corrected

P17: “organic matter content which can be estimated by LOI value”

organic matter influences LOI but cannot be estimated from LOI

Author Response

We thank the helpful comments. We largely agree with the points raised and considered them in the revised version of the manuscript.
In the following pdf, our changes are listed. 

Reviewer 3 Report

Really much work done for this important topic. I see this as a completed work.

Author Response

We appreciate the time you took for our manuscript and the positive feedback provided.
Thank you very much.

Best regards

Round 2

Reviewer 1 Report

The authors mostly made the changes and/or evaluations I suggested. In particular, the mineral contents of the samples examined and other analysis results (chemical composition, SSA, CEC, abrasion, viscosity, consistency, etc.) should be compatible with each other. The authors have made some changes in the text, taking into account some of my criticisms, some explanations and suggestions about mineral contents and other data. However, this table has not been modified/edited for many samples whose calculation of mineral contents (especially given in Table 2a and partially in 2b) are obviously incorrect. For this reason, the compatibility of the mineralogical composition with the results of the chemical analysis is important for the explanation of the other properties determined and forms the basis for the interpretation of the data. The authors stated that they used the peak areas of the basal reflection in the calculation of mineral contents, but the accuracy of this method is quite poor. Therefore, they suggested that it should be evaluated in combination with the results of chemical analysis. This suggestion was ignored. Therefore, detection limits of the mineral content and precision depend of the mineral crystallinity, and poor crystallinity results in peak-brooding. In addition, in clay-containing samples, it is often difficult to accurately determine the mineral contents due to the high absorption coefficients of these minerals, the fact that they are not always oriented with respect to their basal plane, their degree of crystallization and very variable crystal structures.

The total mineral content can be re-calculated using XRD data and also subsequently compared with element oxides refine data.

After the recalculation of the mineral contents by considering the major element oxide results, it will be appropriate to interpret the other analysis results.

Author Response

We thank the helpful comments. We have carefully revised the manuscript:
“The total mineral content can be re-calculated using XRD data and also subsequently compared with 
element oxides refine data.” 
Done, on Table 2a (considering chemical data and the poor crystalline /amorphous oxides/hydroxides and 
sulphates) and on table 2b (recalculating peak areas and correction factors)
“After the recalculation of the mineral contents by considering the major element oxide results, it will be 
appropriate to interpret the other analysis results.” 
It was done.
Sincerely,
The authors
